# FedBug: A Bottom-Up Gradual Unfreezing Framework for Federated Learning with Client Drift

## Abstract

Federated Learning (FL) offers a collaborative training framework, allowing multiple clients to contribute to a shared model without compromising data privacy. Due to the heterogeneous nature of local datasets, updated client models may overfit and diverge from one another, commonly known as the problem of client drift. In this paper, we propose FedBug (Federated Learning with Bottom-Up Gradual Unfreezing), a novel FL framework designed to effectively mitigate client drift. FedBug adaptively leverages the client model parameters, distributed by the server at each global round, as the reference points for cross-client alignment. Specifically, on the client side, FedBug begins by freezing the entire model, then gradually unfreezes the layers, from the input layer to the output layer. This bottom-up approach allows models to train the newly thawed layers to project data into a latent space, wherein the separating hyperplanes remain consistent across all clients. We theoretically analyze FedBug in a novel over-parameterization FL setup, revealing its superior convergence rate compared to FedAvg. Through comprehensive experiments, spanning various datasets, training conditions, and network architectures, we validate the efficacy of FedBug. Our contributions encompass a novel FL framework, theoretical analysis, and empirical validation, demonstrating the wide potential and applicability of FedBug.

## 1 Introduction

Federated Learning (FL) is a distributed approach that enables multiple clients to collaboratively train a shared model without disclosing their raw data. Federated Average (`FedAvg`) (McMahan et al., 2017), one of the most influential FL frameworks, has served as the cornerstone for numerous algorithms in the field. Below, we provide a concise explanation of how `FedAvg` operates: It involves a central server and several clients. In each global round, the server distributes the current model to all clients. Each client independently trains its model using its local data until convergence. Once the local training is completed, the clients send their models back to the server. The server then averages these models to obtain an updated global model, which is subsequently employed in the next round.

In FL, *client drift* refers to the inconsistency between models learned by different clients, arising primarily due to the disparities in their private data distribution (Karimireddy et al., 2020b; Luo et al., 2021; Li et al., 2022; Guo et al., 2022). As local models overfit to their datasets and converge to local minima, the global model — derived from averaging client models — compromises in terms of convergence and performance (Li et al., 2020; Zhao et al., 2018; Zhang et al., 2022). Extensive studies have developed strategies to tackle the client drift issue. We specifically focus on those utilizing *anchors* shared among clients, including gradient anchors and feature anchors. Gradient anchors involve the use of shared gradient information to guide the update of the client model, thereby promoting alignment and mitigating client drift (Karimireddy et al., 2020b; Xu et al., 2021; Das et al., 2022; Karimireddy et al., 2020a; Li et al., 2019). On the other hand, feature anchors rely on shared feature information to assist in feature alignment and regularization of the feature space (Luo et al., 2021; Tang et al., 2022; Tan et al., 2022; Xu et al., 2023). However, these methods may necessitate the extra transmission of gradient information or increased regularization costs (Karimireddy et al., 2020a; Xu et al., 2021; Karimireddy et al., 2020b; Li et al., 2019) and pose privacy concerns (Luo et al., 2021; Tan et al., 2022; Xu et al., 2023).

To mitigate the client drift problem, another line of research has focused on leveraging model's own parameter as anchors. For example, FedBABU (Oh et al., 2021) proposes to fix the classifier throughout training and update it only during evaluation. As all clients share the same fixed classifier, a set of decision boundaries is common to all clients, serving as a shared reference for updating the encoder. While FedBABU yields promising results in personalized FL scenarios, where models are allowed to be fine-tuned using the client's private data during the evaluation stage, FedBABU's performance in general FL settings is less optimal and lacks theoretical understanding.

In this work, we seek to extend FedBABU towards a more robust FL framework, so that improved trainability and FL generalization can be achieved. Our approach hinges on two insights: (1) At the start of each global round of `FedAvg`, all clients receive an identical model from the server, and (2) each intermediate layer parameterizes hyperplanes that separate latent features. Taken together, these insights suggest a strategy: By freezing the models received from the server, we can exploit the consistency of the hyperplanes across clients to provide a common feature space for alignment.

Building on the above insights, we introduce **FedBug** (**Fed**erated Learning with **B**ottom-**U**p **G**radual Unfreezing), a FL framework leveraging shared parameter anchors to mitigate client drift. Unlike `FedAvg`, `FedBug` begins local training by freezing the entire model, then gradually thaws the layers from the input layer to the output layer. The key mechanism operates as follows: when a layer becomes trainable (thawed) while its succeeding layers remain frozen, this thawed layer learns to project its inputs into a shared feature space. This space is notably defined by the hyperplanes of the still-frozen succeeding layers, providing a common reference across clients and enhancing cross-client alignment. Then, `FedBug` progressively unfreezes the next layer, ensuring the layer's trainability after it has served as a shared reference while balancing alignment and adaptability. As detailed later, we investigate `FedBug` through both theoretical analysis and empirical experiments. The bottom-up unfreezing strategy is shown to efficiently fit data distributions across clients. Thus, parameter updates can be performed to tackle the client drift issue. We further quantify such influence and reveal `FedBug` converges faster than `FedAvg`. Additionally, we conduct experiments across various datasets, training conditions, and architectures to ensure the broad applicability of `FedBug`.

Our contributions are three-fold:

- **Novel Federated Learning Framework:** We propose `FedBug`, a unique federated learning framework that leverages model parameters as anchors to effectively address the challenges of client drift in federated learning.

- **Theoretical Analysis:** We provide a theoretical analysis of the `FedBug` framework, demonstrating its better convergence rate compared to `FedAvg`. Our analysis focuses on a one-layer linear network with an orthogonal regression task, offering novel insights into federated learning dynamics in the context of over-parameterized models.

- **Empirical Validation:** We extensively validate the effectiveness of `FedBug` through a series of experiments on diverse datasets (CIFAR-10, CIFAR-100, Tiny-ImageNet), varying training conditions (label skewness, client participation rate), and different model architectures (standard CNN, ResNet18, ResNet34). Furthermore, we assess the compatibility of the `FedBug` framework with other federated learning algorithms. Our empirical findings underscore the immense potential and wide-ranging applicability of `FedBug`.

## 2 LITERATURE REVIEW

**Mitigating Client Drift Using Gradient and Feature Anchors.** To address the client drift problem in federated learning, various methods have been explored to explicitly or implicitly provide shared reference points for client models. **Gradient anchors** are employed in several FL methods. For instance, SCAFFOLD (Karimireddy et al., 2020b) incorporates server-level gradients in updates to reduce local noise effects, while FedCM (Xu et al., 2021), FedGLOMO (Das et al., 2022), MIME (Karimireddy et al., 2020a), and FedDANE (Li et al., 2019) leverage server-level gradient to align clients by providing mutual update directions. While these approaches have demonstrated effectiveness, they require sharing additional gradient information with the local clients. **Feature anchors** are used in methods such as CCVR (Luo et al., 2021), VHL (Tang et al., 2022), FedProto (Tan et al., 2022), and FedFA (Xu et al., 2023), employing outside datasets or clients' private datasets to

Figure 1: **Comparisons of `FedAvg` and `FedBug` on the Client Side.** While `FedAvg` updates all network modules during local training on the client side, `FedBug` strategically employs a bottom-up approach to gradually unfreeze network modules, aiming to counteract client drift. Take `FedBug` (40%) for example, the first 40% of the local iterations perform gradual unfreezing (GU), while the remaining 60% perform vanilla training. Despite maintaining the same number of training iterations as `FedAvg`, `FedBug` updates fewer parameters, leading to enhanced learning efficiency.

produce and regularize features across different clients. However, these approaches may suffer from privacy leakage, increased dataset or feature transmission costs, and added computation budget.

**Mitigating Client Drift Using Parameter Anchor.** Recent research has emphasized the prominence of client drift in the top layers of models. Specifically, it has been shown that the penultimate layer and classifier exhibit the lowest feature similarities among the clients (Zhao et al., 2018; Luo et al., 2021; Li et al., 2022; Guo et al., 2022). These findings suggest that local classifiers undergo significant changes to adapt to the local data distribution, which amplifies challenges related to class imbalance and results in biased model predictions for specific classes (Zhang et al., 2022; Shang et al., 2022; Lee et al., 2021; Guo et al., 2022). This phenomenon is consistent with research on the long-tail analysis, where Yu et al. (2020) and Kang et al. (2020) demonstrated that the head is biased in class-imbalanced environments. To explicitly address the issue of client drift, certain studies leverage parameter anchors to align the clients. FedProx (Li et al., 2020) regularizes the L2 distance between the client model and server model, establishing a shared reference in the parameter space. Additionally, FedBABU (Oh et al., 2021) falls into this category as it uses a fixed classifier as the parameter-level anchor during training. As each client's frozen classifier parameterizes the same set of decision boundaries in the feature space, FedBABU also serves as a hyperplane anchor in the latent space. However, FedBABU's performance is suboptimal and necessitates additional personalized training (fine-tuning) on the clients' private dataset during the evaluation stage, a process known as personalization, to achieve satisfactory results.

## 3 METHOD

The core principle of `FedBug` is to mitigate client drift by strategically controlling the update of model parameters. As detailed in Figure 1, `FedBug` divides the local training phase into two stages. In the **gradual unfreezing (GU) stage**, the entire model is initially frozen (i.e., cannot be updated). Over time, modules are progressively unfrozen, starting from the bottom layers upwards. This stage's duration is broken into periods, and a new module becomes trainable at the start of each period. Afterward, in the **vanilla training stage**, the entire model becomes trainable, allowing updates across all layers. Notably, the number of training iterations of `FedBug` matches that of `FedAvg`.

We first introduce the notation used in this work. The model is represented as $\theta$, comprising $m$ modules. Here, a *module* can refer to a single CNN layer or a ResNet block. For simplicity, we use $\theta^{1:m}$ to denote the first $m$ modules of the model, with $\theta^1$ indicating the input module. We define $P$ as the fraction of the GU stage and $K$ as the total number of local training iterations. Thus, the GU stage spans $PK$ iterations. We denote $\eta_g$ and $\eta_l$ as global and local learning rates, respectively. Algorithm 1 details the `FedBug` framework. In line seven of the algorithm, the variable $m$ is determined based on the current local iteration step $k$, indicating which modules should be updated. For a comprehensive comparison between `FedAvg` and `FedBug`, please refer to Appendix C.

We now explore the underlying rationale of `FedBug`, using Figure 1 for illustration. Consider a four-module model trained using `FedBug` (40%), represented as $M = 4$, and $P = 0.4$. In this case, a module becomes trainable every $0.1K$ local iteration. Suppose we are in the second GU period, where all clients have just unfrozen their second module. During this period, the clients adapt their first and second modules and project the data into a feature space. Notably, the separating hyperplanes within this feature space are parameterized by the yet-to-be-unfrozen modules (the third and fourth modules in this case). These modules remain consistent during this period, serving as a shared anchor among clients. Similarly, as we progress to the subsequent third period, this process continues, with clients mapping their data into decision regions defined by the still-frozen fourth module. By leveraging the shared reference, `FedBug` ensures ongoing alignment among the clients.

---

**Algorithm 1** `FedBug`

**Notation**:
$\theta^{1:m}$: the first $m$ modules of model $\theta$
$R$: number of global rounds
$K$: number of local iterations
$P$: gradual unfreezing stage percentage

1: **Input**: global model $\theta$ with $M$ modules
2: **for** $r = 1, \ldots, R$ **do**
3:     Sample clients $S \subseteq \{1, ..., N\}$
4:     **for** each client $i \in S$ in parallel **do**
5:         Initialize local model $\theta_i \leftarrow \theta$
6:         **for** $k = 1, \ldots, K$ **do**
7:             $m \leftarrow \min\{M, \lceil \frac{kM}{PK} \rceil\}$
8:             // Update $m$ modules of $\theta_i$
9:             $\theta_i^{1:m} \leftarrow \theta_i^{1:m} - \eta_l \nabla F_i(\theta_i^{1:m})$
10:         **end for**
11:         $\Delta_i \leftarrow \theta_i - \theta$
12:     **end for**
13:     $\theta \leftarrow \theta + \frac{\eta_g}{|S|} \sum_{i \in \mathcal{S}} \Delta_i$
14: **end for**

---

## 4   THEORETICAL ANALYSIS

In this section, we provide theoretical analysis for the learning behaviors of `FedAvg` and `FedBug`. Our goal is to show that `FedBug` exhibits improved convergence than `FedAvg` while better handling data with client drift.

### 4.1   TASK SETTING AND MODEL ARCHITECTURE

**Task and Evaluation.** For simplicity, we consider a FL regression task with two clients denoted as $c_1$ and $c_2$. Each client has different regression data, specifically $\mathcal{T}_1 = \{x_1 = [1, 0], y_1 = 1\}$ and $\mathcal{T}_2 = \{x_2 = [0, 1], y_2 = 1\}$. The objective is to minimize the L2 loss, with client $c_1$ ($c_2$) minimizing $L_1 = \|f(x_1) - y_1\|$ ($L_2 = \|f(x_2) - y_2\|^2$), where $f$ denotes the model.

**Model Architecture.** We start from the model architecture of a one-layer linear network $f$ with two nodes $[a, b]$ (as model weights) and a bias term $v$. Formally, the function is described as $f(x) = x[a, b]^\top + v$. In this setup, the task setup implies client $c_1$ ($c_2$) aims to minimize $L_1 = |a + v - 1|^2$ ($L_2 = |b + v - 1|^2$). A global solution must satisfy $L_1 = L_2 = 0$, indicating $a = b$.

In this setup, we employ three variables—weights $(a, b)$ and bias $v$—to represent the multi-layered relationships inherent in a neural model. Specifically, the input layers directly interface with diverse data distributions, whereas the top layers interact indirectly, primarily engaging with the outputs from the bottom layers. Our analysis delves into the logic behind `FedBug`'s bottom-up unfreezing strategy: If the bottom layers, represented by $a$ and $b$, can adeptly adapt to the dataset distribution, then using the subsequent layers to co-adjust to data distribution could inadvertently induce overfitting and amplify unnecessary parameter updates, leading to the client drift issue.

### 4.2   PRELIMINARY: FEDAVG AND FEDBUG

We review `FedAvg` and its notations. During the $i$-th global round, the server distributes parameters $[a^i, b^i, v^i]$ to the clients. For example, client $c_1$ receives the initial parameter $[a_{c_1,0}^i, b_{c_1,0}^i, v_{c_1,0}^i](= [a^i, b^i, v^i])$. Each client individually optimizes cost function using their own parameters. After $k$ local iterations, the parameters of client $c_1$ are updated to $[a_{c_1,k}^i, b_{c_1,k}^i, v_{c_1,k}^i]$, and upon achieving local convergence, they become $[a_{c_1,*}^i, b_{c_1,*}^i, v_{c_1,*}^i]$. Once local convergence is reached, clients send their learned parameters back to the server. The server then averages the received parameters to obtain the new parameters $[a^{i+1}, b^{i+1}, v^{i+1}] = \frac{1}{2}([a_{c_1,*}^i, b_{c_1,*}^i, v_{c_1,*}^i] + [a_{c_2,*}^i, b_{c_2,*}^i, v_{c_2,*}^i])$.

`FedBug` differs from `FedAvg` only in the client-side update step: Clients first freeze $v$ and update $[a, b]$ for $N_{\text{step}}$ local iterations. Afterward, the client unfreezes the last layer parameters and performs gradient descent on all the parameters. In this section, we assume $N_{\text{step}} = 1$.

With the above task, we are able to assess the associated algorithmic convergence. Instead of primarily focusing on the speed of loss reduction towards a global minimum, we can design a surrogate metric. At global convergence, the condition $a^* = b^*$ must hold given $a^* + v^* = 1$ and $b^* + v^* = 1$. Thus, the distance between $a^i$ and $b^i$ in the $i$-th global round serves as an indicator of its deviation from the global minima. Furthermore, to understand the rate of this deviation contraction, we evaluate the client discrepancy scaling. This is depicted through the subsequent definitions:

**Definition 1.** *Client discrepancy $d^i$ is the L1 distance between the server model parameters $a^i$ and $b^i$ at the $i$-th global round : $d^i = \|a^i - b^i\|$.*

**Definition 2.** *Client discrepancy contraction ratio $r$: $r = d^{i+1}/d^i$.*

### 4.3 THE CONVERGENCE RATE OF FEDAVG AND FEDBUG

We now investigate the convergence rates of `FedAvg` and `FedBug`. We aim to highlight the enhanced convergence of `FedBug` compared to `FedAvg`. We begin by presenting our theoretical findings regarding the convergence behavior of `FedAvg`.

**Theorem 1.** *`FedAvg` converges with a client discrepancy contraction ratio of $\frac{3}{4}$.*

*Proof.* Run `FedAvg` until reaching local minima, the parameter of client $c_1$ is $(a^i_{c_1,*}, b^i_{c_1,*}, v^i_{c_1,*}) = (\frac{1+(a^i-v^i)}{2}, b^i, \frac{1-(a^i-v^i)}{2})$, while that of client $c_2$ is $(a^i_{c_2,*}, b^i_{c_2,*}, v^i_{c_2,*}) = (a_i, \frac{1+(b^i-v^i)}{2}, \frac{1-(b^i-v^i)}{2})$. This is obtained because the gradient of client $c_1$ follows $a^i_{c_1,k} - v^i_{c_1,k} = a^i - v^i$ for each $k$ local iteration and that the minima satisfied $a^i_{c_1,*} + v^i_{c_1,*} = 1$, similarly for client $c_2$.

Therefore, we have $d^{i+1} = \|\frac{a^i+a^i_{c_1,*}}{2} - \frac{b^i+b^i_{c_2,*}}{2}\| = \frac{1}{2}d^i + \frac{1}{2}\|\frac{1}{2}(a^i - b^i)\| = \frac{3}{4}d^i$.

This yields the ratio $r = \frac{d^{i+1}}{d^i} = \frac{3}{4}$. $\qquad\square$

**Theorem 2.** *With a step size $\eta < 1$ during the gradient unfreezing stage, `FedBug` converges with a client discrepancy contraction ratio $\frac{3-\eta}{4}$.*

*Proof.* To validate Theorem 2, we divide the learning process into two stages. In stage one, we update the parameters $a^i_{c1,0}$ and $b^i_{c2,0}$ for one step, while freezing the last layer parameter $v$. In the succeeding stage, the parameter $v$ is unlocked, and gradient descent is applied across all parameters until each client reaches its local minimum.

In the first stage, we obtain $a^i_{c1,1} = a^i - \eta(a^i + v^i - 1)$ and $b^i_{c2,1} = b^i - \eta(b^i + v^i - 1)$.

In the second stage, the model parameter of client $c_1$ reaches $(a^i_{c_1,*}, b^i_{c_1,*}, v^i_{c_1,*}) = (\frac{1+(a^i_{c1,1}-v^i_{c1,1})}{2}, b^i, \frac{1-(a^i_{c1,1}-v^i_{c1,1})}{2})$, and that of client $c_2$ reaches $(a^i_{c_2,*}, b^i_{c_2,*}, v^i_{c_2,*}) = (a_i, \frac{1+(b^i_{c1,1}-v^i_{c1,1})}{2}, \frac{1-(b^i_{c1,1}-v^i_{c1,1})}{2})$. This results in:

$$d^{i+1} = \frac{1}{2}d^i + \frac{1}{2}\|\frac{1}{2}(a^i_{c1,1} - b^i_{c2,1})\| = \frac{1}{2}d^i + \frac{1}{4}\|(1-\eta)(a^i - b^i)\| = \frac{3-\eta}{4}d^i$$

Thus, yielding the ratio $r = \frac{d^{i+1}}{d^i} = \frac{3-\eta}{4}$. $\qquad\square$

**Insights.** Given the dataset setup, the input nodes encounter distinct dataset domains directly. If these input nodes quickly align with their respective dataset domains, the need to adapt the bias term becomes minimal. However, with `FedAvg`, each client updates all parameters simultaneously, adjusting the bias term to achieve its local minima. This gives rise to a distinct challenge: The bias term, originally intended as a shared resource, becomes a tool for each client to achieve a tighter fit to its dataset, thus destabilizing this shared term. As showed in the proof for theorem 1, where $\|a^i_{c_1,*} - b^i_{c_2,*}\| = \|\frac{1+(a^i-v^i)}{2} - \frac{1+(b^i-v^i)}{2}\| = \|\frac{a^i-b^i}{2}\|$, the effect of $v^i$ gets neutralized. This implies that even though clients are exploiting the bias term for fitting, its impact becomes null from a global

perspective. In contrast, `FedBug` addresses this by freezing the bias term for one local iteration, ensuring both clients update their own nodes in a manner beneficial to all.

**Connection to `FedBug`.** The rationale behind `FedBug`'s bottom-up unfreezing approach is underscored by the behavior of multi-layer neural networks: If the initial layers alone can effectively capture the dataset distribution, then having the subsequent layers to co-adapt to the data distribution may inadvertently promote overfitting. Furthermore, such co-adaptation results in more pronounced parameter shifts, amplifying disparities between clients. This understanding becomes clear when observing that in multi-layer neural networks, the initial layers interact with diverse dataset distributions, similar to the weight nodes $[a, b]$. Conversely, the top layers more closely resemble the bias term, as they do not directly engage with the input distribution.

**Generalization of Analysis.** Our analytical framework is adaptable to a multiple clients scenario. This detailed discussion, along with the revised definition of Client Discrepancy and considerations on multi-layered models, can be found in Appendix B.

## 5 EXPERIMENTS

We present an extensive evaluation of `FedBug` across various FL scenarios. We provide a brief overview of the datasets and models, with a detailed description of our setup available in the Appendix. The code is written in PyTorch and executed on a single GPU, either an NVIDIA 3090 or V100. All experiments are performed with four distinct random seeds. Experimental details along with the ablation study are deferred to Appendices A.1 and A.2, respectively.

### 5.1 EXPERIMENTAL SETUP

**Datasets.** We utilized benchmark datasets following the same train/test splits as previous works (McMahan et al., 2017; Li et al., 2020; Acar et al., 2021). These datasets include CIFAR-10, CIFAR-100, and Tiny-ImageNet. We randomly assigned data to the clients for the IID label distribution split (McMahan et al., 2017; Acar et al., 2021). As for the non-IID label distribution, we followed the Dirichlet distribution $\text{Dir}(\alpha)$, as in Yurochkin et al. (2019) and Acar et al. (2021). Here, $\alpha$ is a concentration parameter, with a smaller $\alpha$ indicating stronger data heterogeneity. When $\alpha$ equals $\infty$, the setting is homogeneous. We set $\alpha$ to $0.3$ for CIFAR-10 and CIFAR-100, and $0.5$ for Tiny-ImageNet (McMahan et al., 2017; Li et al., 2020; Acar et al., 2021).

**Models.** For standard CNN, we employ a standard convolutional neural network, similar to McMahan et al. (2017) and Acar et al. (2021), consisting of two (three) convolutional layers followed by three fully connected layers for CIFAR-10 and CIFAR-100 (Tiny-ImageNet) dataset. For ResNet-18 and ResNet-34 (He et al., 2016), we change the batch normalization to group normalization (Acar et al., 2021; Hosseini et al., 2021; Yu et al., 2021; Hyeon-Woo et al., 2021; Hsieh et al., 2020).

### 5.2 EXPERIMENTAL RESULTS

**Improved Performance and High Compatibility With Various FL Algorithms.** The `FedBug` algorithm seamlessly integrates with numerous existing FL algorithms. For instance, to combine `FedBug` with FedDyn, one simply needs to add the regularization term of FedDyn to the original loss, leaving our defined gradual unfreezing schedule untouched. Results from CIFAR-100 and Tiny-ImageNet are shown in Figures 2 and 3, respectively. Results on CIFAR-10 can be found in Figure 7 within the Appendix. These results highlight the high compatibility of the `FedBug` framework when combined with different FL algorithms. Additionally, we consistently observe that `FedBug` outperforms the vanilla training framework, even when the gradual unfreezing stage comprises only ten percent of the local training process. This indicates the effectiveness and efficiency of the `FedBug` approach in improving model performance. Furthermore, our `FedBug` training framework exhibits a consistent synergistic effect across five distinct FL algorithms, two client participation levels, and both IID and non-IID label distributions. This demonstrates the broad applicability of our proposed framework in combination with existing FL training algorithms and experimental setups.

**Applicability of `FedBug` on ResNet.** When applying `FedBug` to larger models like ResNet, a natural question arises: What should be the smallest unit to unfreeze during the GU stage — a ResNet Module or a residual block? Since both ResNet-18 and ResNet-34 can be seen as having four ResNet

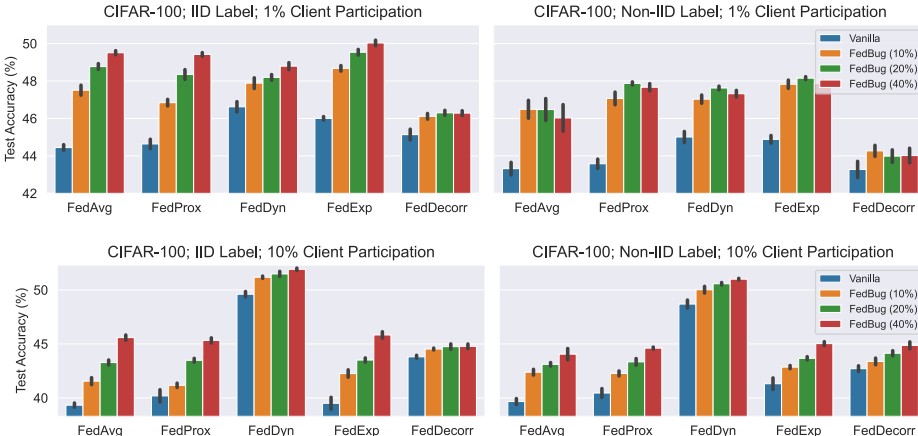

Figure 2: **Experiments on CIFAR-100 with standard CNN.** We conduct experiments at different client participation rates (1% and 10%), levels of heterogeneity ($\alpha \in \{0.3, \infty\}$), and combinations of FL algorithms. Results are averaged across four seeds, and the error bar indicates deviation.

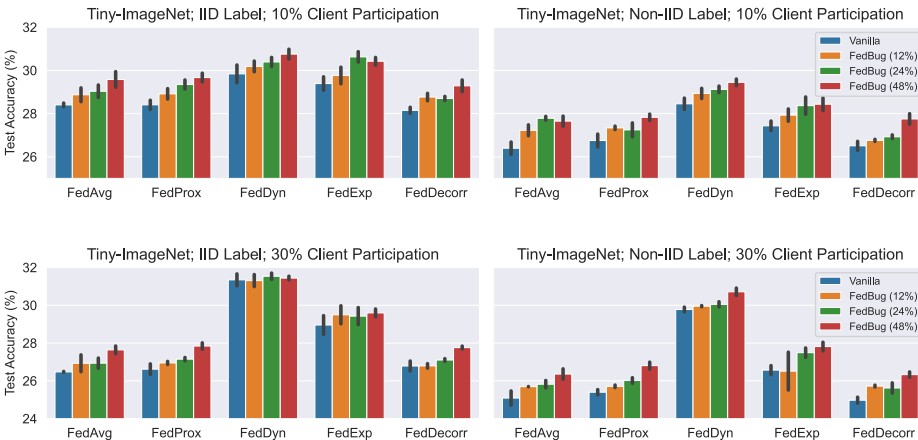

Figure 3: **Experiments on Tiny-ImageNet with** 10 **clients with standard CNN.** We conduct experiments at different client participation rates (10% and 30%), levels of heterogeneity ($\alpha \in \{0.3, \infty\}$), and combinations of FL algorithms.

Modules or consisting of eight and sixteen residual blocks, respectively, a robust framework should remain agnostic to the unit definition. Therefore, we evaluate the adaptability of `FedBug` using two distinct unfreezing strategies: (1) progressively unfreezing one ResNet Module at a time, and (2) progressively unfreezing one residual block at a time. Notably, we capitalize the term ResNet "Module" to differentiate it from the general module mentioned in Algorithm 1.

The experimental results, presented in Table 2, consistently demonstrate the superiority of the `FedBug` framework over the vanilla training framework across both unfreezing strategies and different label distributions. Interestingly, we observe that both strategies perform comparably well on the datasets, indicating the effectiveness of `FedBug`.

**Impact of Gradual Unfreezing Percentage.** We investigate the impact of the percentage of the GU stage in CIFAR-100 and Tiny-ImageNet in the standard CNN model. The baseline framework is `FedAvg`, represented by a percentage of $0\%$. The results are shown in Figure 4. Our experiments reveal consistent improvements in test accuracy even with longer GU ratios. Notably, allocating a larger percentage of the training period to GU leads to the top layers receiving less training. Thus, the test accuracy does not necessarily increase monotonically with the GU ratio. For instance, when training a five-layer model with `FedBug` using a 100% GU stage ratio, the penultimate linear layer and the classifier are trained for only 40% and 20% of the total training iterations, respectively. In this

| | Tiny-ImageNet (# clients: 10; client participation rate: **10%**) | | | | | | | | | |
|---|---|---|---|---|---|---|---|---|---|---|
| Method | IID label distribution ($\alpha = \infty$) | | | | | Non-IID label distribution ($\alpha = 0.5$) | | | | |
| | FedAvg | FedProx | FedDyn | FedExp | FedDecorr | FedAvg | FedProx | FedDyn | FedExp | FedDecorr |
| Vanilla | 26.48 | 26.62 | 31.35 | 26.79 | 28.96 | 25.09 | 25.40 | 29.78 | 24.98 | 26.57 |
| FedBug (12%) | 26.93 | 26.95 | 31.31 | 26.79 | 29.50 | 25.69 | 25.71 | 29.95 | 25.72 | 26.52 |
| FedBug (24%) | 26.94 | 27.15 | **31.54** | 27.10 | 29.43 | 25.82 | 26.02 | 30.05 | 25.62 | 27.49 |
| FedBug (48%) | **27.64** | **27.84** | 31.44 | **27.76** | **29.60** | **26.36** | **26.81** | **30.71** | **26.33** | **27.82** |

| | Tiny-ImageNet (# clients: 10; client participation rate: **30%**) | | | | | | | | | |
|---|---|---|---|---|---|---|---|---|---|---|
| Method | IID label distribution ($\alpha = \infty$) | | | | | Non-IID label distribution ($\alpha = 0.5$) | | | | |
| | FedAvg | FedProx | FedDyn | FedExp | FedDecorr | FedAvg | FedProx | FedDyn | FedExp | FedDecorr |
| Vanilla | 28.40 | 28.41 | 29.84 | 29.39 | 28.15 | 26.40 | 26.76 | 28.46 | 27.44 | 26.51 |
| FedBug (12%) | 28.87 | 28.92 | 30.18 | 29.76 | 28.77 | 27.23 | 27.34 | 28.94 | 27.93 | 26.76 |
| FedBug (24%) | 29.04 | 29.35 | 30.39 | **30.63** | 28.70 | **30.76** | 27.25 | 29.12 | 28.37 | 26.93 |
| FedBug (48%) | **29.58** | **29.67** | 27.78 | 30.42 | **29.29** | 27.65 | **27.83** | **29.45** | **28.44** | **27.75** |

Table 1: **Experiments on Tiny-ImageNet of** 10 **clients with standard CNN.** We conduct experiments with different client participation rates (10% and 30%), degrees of heterogeneity ($\alpha \in \{0.3, \infty\}$), and combinations of FL algorithms.

| | CIFAR-100 (# clients: 10; client participation rate: 10%) | | | | | | | |
|---|---|---|---|---|---|---|---|---|
| Method | IID label Distribution ($\alpha = \infty$) | | | | Non-IID label Distribution ($\alpha = 0.3$) | | | |
| | ResNet-18 | | ResNet-34 | | ResNet-18 | | ResNet-34 | |
| | Module(4) | Block(8) | Module(4) | Block(16) | Module(4) | Block(8) | Module(4) | Block(16) |
| Vanilla | 52.59 | 52.59 | 52.64 | 52.64 | 49.04 | 49.04 | 48.69 | 48.69 |
| FedBug (20%) | 53.25 | 53.05 | 53.01 | 53.42 | **49.70** | 49.64 | 49.20 | 49.17 |
| FedBug (40%) | **53.65** | **53.49** | **53.56** | **53.56** | 49.36 | **49.69** | **49.37** | **49.33** |

| | Tiny-ImageNet (# clients: 10; client participation rate: 10%) | | | | | | | |
|---|---|---|---|---|---|---|---|---|
| Method | IID label Distribution ($\alpha = \infty$) | | | | Non-IID label Distribution ($\alpha = 0.5$) | | | |
| | ResNet-18 | | ResNet-34 | | ResNet-18 | | ResNet-34 | |
| | Module(4) | Block(8) | Module(4) | Block(16) | Module(4) | Block(8) | Module(4) | Block(16) |
| Vanilla | 33.88 | 33.88 | 33.22 | 33.22 | 31.91 | 31.91 | 31.53 | 31.53 |
| FedBug (20%) | 34.25 | 34.31 | 34.28 | 34.36 | 32.29 | 32.32 | 32.33 | 32.31 |
| FedBug (40%) | **35.28** | **35.17** | **35.12** | **35.10** | **32.86** | **33.47** | **33.20** | **33.36** |

Table 2: **Experiments on ResNet with module-wise and block-wise unfreezing strategies.** Note that two unfreezing strategies are considered: (1) unfreezing one ResNet *Module* at a time and (2) unfreezing one residual *Block* at a time. Module (4) indicates that the model consists of four ResNet Modules, while Block (16) signifies the model consists of sixteen residual Blocks. Consistent improvements of `FedBug` with both unfreezing strategies can be observed.

extreme scenario, `FedBug` not only saves considerable training time but also provides improvement. These results highlight the robustness of `FedBug` to the GU ratio and suggest a small portion of the training time for gradual unfreezing may readily yield favorable results.

**Ablation Study: Comparative Analysis of Freezing Strategies.** To assess the impact of different freezing strategies, we consider the following closely related methods: (1) Top-Down Gradual Unfreezing, which is used in recent NLP literature for model fine-tuning (Howard & Ruder, 2018; Mukherjee & Awadallah, 2019; Raffel et al., 2020; Liu et al., 2023). This approach fine-tunes the model from the output layer to the input layer. (2) Fixing the last layer throughout training, known as FedBABU (Oh et al., 2021); and (3) Fixing the last two Layers, i.e., the classifier and the penultimate layer. Our baseline strategies are `FedAvg` and `FedBug` (20%), referred to as "Vanilla" and "FedBug: Bottom-Up GU", respectively.

The results for CIFAR-10, CIFAR-100, and Tiny-ImageNet are depicted in Figure 5. These results indicate that all alternative unfreezing or freezing methods underperform when compared to `FedAvg`. In contrast, our proposed `FedBug` strategy consistently outperforms `FedAvg`, underscoring the significance of the bottom-up sequential order for unfreezing in achieving optimal test performance.

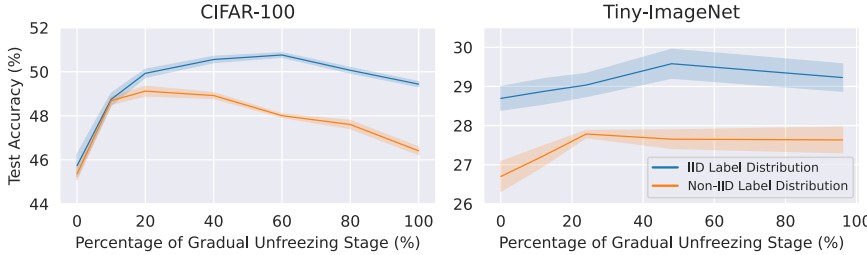

Figure 4: **Impact of Gradual Unfreezing Percentage.** We investigate the effect of the percentage of the GU stage in CIFAR-100 and Tiny-ImageNet. The baseline framework is `FedAvg` (i.e., GU percentage of 0.0). Our experiments reveal consistent improvements in test accuracy even with longer GU ratios, where the top layers are relatively under-trained compared to `FedAvg`.

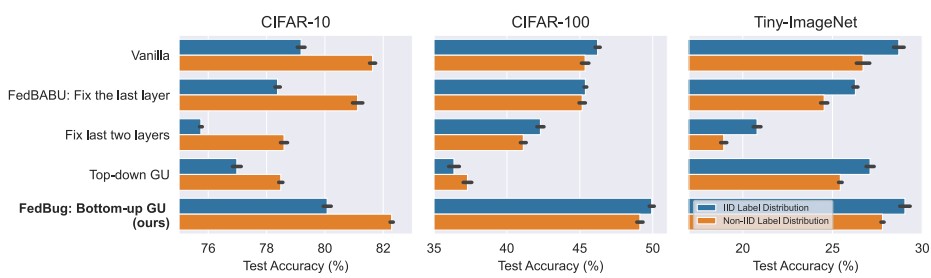

Figure 5: **Comparative Analysis of Freezing Strategies.** `FedBug` consistently outperforms other freezing strategies across three datasets, including `FedBABU` of Oh et al. (2021).

**Reduced Running Clock Time with `FedBug`.** Despite both `FedBug` and `FedAvg` using equal number of local iterations in the client side, `FedBug` showcases enhanced training efficiency. This is attributed to its unique freezing-unfreezing mechanism which updates fewer parameters, thus hastening early local iterations. We measured the clock time needed for gradient computations and parameter updates on individual batches (using Pytorch 1.12 on NVIDIA GeForce RTX 3090), deploying a standard CNN model on CIFAR-100 with a 1% client participation rate. Figure 6 presents these results, averaged over 10,000 batches. Remarkably, the `FedBug` design significantly expedites batch computation time. A GU ratio of 80% achieves a notable acceleration of 119% per batch. Even at modest GU percentages, the speeds of 102% and 105% are noteworthy, leading to substantial time and resource savings in extensive experiments.

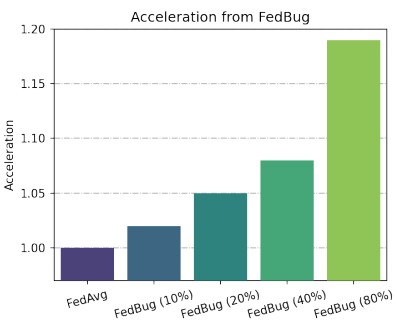

Figure 6: Batch computation time acceleration using `FedBug` with varying Gradual Unfreezing ratios.

## 6 CONCLUSION

In this work, we introduce `FedBug`, a novel FL framework designed to mitigate client drift. By leveraging model parameters as anchors, `FedBug` aligns clients while improving learning efficiency. We perform theoretical analysis in an over-parameterized setting, revealing that `FedBug` achieves a faster convergence rate compared to the widely adopted `FedAvg` framework. To empirically validate the effectiveness, we conduct extensive experiments on various datasets, training conditions, and network architectures, consistently demonstrating the superiority and compatibility of `FedBug`. Overall, our contributions include the introduction of a novel FL framework, theoretical analysis, and comprehensive empirical validation, highlighting the broad potential and applicability of FedBug.

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
