# A    EXPERIMENTS

## A.1    IMPLEMENTATION DETAILS

**Standard CNN.** Our code is based on Acar et al. (2021) and we extend it to include the Tiny-ImageNet dataset, the latest proposed FedDecorr (Shi et al., 2022) and FedExP (Jhunjhunwala et al., 2022) algorithms. We use Stochastic Gradient Descent (SGD) optimizer and a cross entropy loss function, with a learning rate of 0.1 and weight decay of 0.001. We use a batch size of 50 and perform horizontal flipping for training data augmentation on all datasets, while adding cropping augmentation on CIFAR-10 and CIFAR-100 (Acar et al., 2021). For the training epochs, we run 300, 500, and 100 global rounds (communication rounds) with 10, 5, and 5 local epochs in CIFAR-10, CIFAR-100, and Tiny-ImageNet, respectively, as suggested in Acar et al. (2021) and shi2022towards. In the CIFAR-10 and CIFAR-100 (Tiny-ImageNet) datasets, we consider 100 (10) participants with client participation rates of 1% and 10% (10% and 30%) at each global round, respectively (Acar et al., 2021). 1% client participation rate means that each client had a 1% chance of being selected to join in a global round.

**ResNet.** For the ResNet architecture, we focus on CIFAR-100 and Tiny-ImageNet datasets. We conducted 100 global rounds with 5 local epochs on CIFAR-100 and 3 local epochs on Tiny-ImageNet (Acar et al., 2021; Shi et al., 2022). In both datasets, we work with 10 participants with a 10% client participation rate. Notably, ResNet-18 and ResNet-34 can be treated as having either four modules each, or eight and sixteen residual blocks, respectively. To test the applicability of FedBug, we consider two strategies: (1) unfreezing one ResNet module at a time, or (2) unfreezing one residual block at a time. The first strategy corresponds to a scenario where $M = 4$, while the second corresponds to $M = 8$ for ResNet-18 and $M = 16$ for ResNet-34.

**Hyperparameters.** Our use of hyperparameters is similar to (Acar et al., 2021), where $\mu = 0.0001$ for FedProx (Li et al., 2020), $alpha = 0.01$ for FedDyn (Acar et al., 2021). We use $\beta = 0.01$ for FedDecorr (Shi et al., 2022), while FedExp (Jhunjhunwala et al., 2022) does not require additional hyperparameters.

## A.2    EXPERIMENTAL RESULTS

**Improved Performance in CIFAR-10.** Similar to experiments on CIFAR-100 and Tiny-ImageNet datasets, FedBug consistently augments baseline algorithms over various client participation rates and levels of heterogeneity.

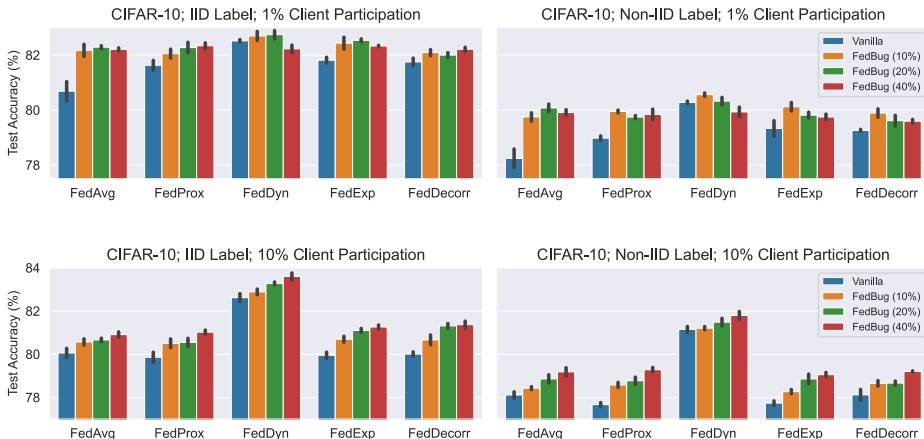

Figure 7: **Experiments on CIFAR-10 of** 10 **clients with standard CNN.** We conduct experiments with different client participation rates (1% and 10%), levels of heterogeneity ($\alpha \in \{0.3, \infty\}$), and combinations of FL algorithms.

**Ablation Study: Number of Clients.** To compare the impact of different numbers of clients, we conducted an ablation study using a consistent client participation rate of 10% for each setting. We

| Method | CIFAR-100 (ResNet-18; client participation rate: 10%) | | | | | |
| | IID label distribution ($\alpha = \infty$) | | | Non-IID label distribution ($\alpha = 0.3$) | | |
| | # Clients | | | # Clients | | |
| | 10 | 50 | 500 | 10 | 50 | 500 |
| Vanilla | 52.55 | 42.40 | 19.45 | 49.05 | 40.65 | 19.32 |
| FedBug | **53.59** | **44.44** | **21.79** | **49.93** | **41.32** | **19.56** |

Table 3: **Experiments on CIFAR-100 with varying number of clients on ResNet-18.**

utilized the CIFAR-100 dataset with ResNet-18 and employed the ResNet Module-wise unfreezing strategy. We utilize `FedBug` (50%) for this ablation study. The experimental results were averaged over four random seeds. The results are summarized in Table 3, revealing that even with a large number of clients, the `FedBug` framework consistently improves testing accuracy.

## B  GENERALIZATION OF THEORETICAL ANALYSIS

**Generalization to Multiple Clients.** Here, we outline how our analysis framework can be extended to accommodate multiple clients and the consideration of more layers.

Firstly, we illustrate a meaningful extension from the case of two clients to a scenario involving multiple clients ($m$ clients), denoted as $c_1, c_2, ..., c_m$, in an FL regression task. Each client's unique regression data $T_i = \{x_i, y_i = 1\}$ for $i = 1, ..., m$, where $x_i$ has a 1 in its $i$-th entry while the other entries are zeros. In this context, adapting our approach entails employing a one-layer linear network with $m$ nodes $[n_1, n_2, ..., n_m]$ in the layer and a single node $[v]$ as the bias term.

To enable this extension, we extend the definition of Client Discrepancy (initially presented in Section 4.3) as follows: "Definition R.1: Client discrepancy $d^i$ is the L1 distance between the server model parameters at the $i$-th global round: $d^i = \sum_{j \neq k} |n_j^i - n_k^i|$." Notably, this adjustment maintains the compatibility of our existing assumptions, thereby ensuring the preservation of the same theorems with minimal alterations. Consequently, our proposed generalization transcends the boundaries of the two-client scenario and holds relevance across broader FL contexts.

**Discussion on Multi-Layered Model.** As for extension to models with more linear layers, we note that our utilization of a one-layer network readily exhibits an over-parameterized scenario, enabling the exploration of the gradual unfreezing benefit. Since a network with multiple layers remains within an overparameterized setting, it may not provide additional theoretical insights. However, we recognize that such extensions involving multiple layers are valuable and may uncover additional effects. The pursuit of such a more complex version of theoretical support to the FL strategy would be among our future directions.

**Discussion on Orthogonal Task Setup.** We provide a direct generalization of the orthogonal task setup, where we use $\mathcal{T}_1 = \{x_1 = e_1, y_1 = 1\}$ and $\mathcal{T}_2 = \{x_2 = e_2, y_2 = 1\}$ as datasets. Here, $e_1$ and $e_2$ are orthogonal vectors, and we can define the model function as $f(x) = x^\top (ae_1 + be_2) + v$.

## C    COMPARISON BETWEEN FEDAVG AND FEDBUG

We provide self-contained outlines of the `FedAvg` and `FedBug` algorithms in Algorithm 2 and Algorithm 3, respectively. The key difference, highlighted in red and blue, is that while `FedAvg` updates all $M$ modules at each local iteration, `FedBug` unfreezes and updates one module at the beginning, progressively training an additional module every $\frac{PK}{M}$ local iterations. As `FedBug` does not require extra information like gradients, momentum, or regularization, it can be easily incorporated into other FL algorithms.

---

**Algorithm 2** `FedAvg`

    **Notation**:
        $\theta^{1:m}$: the first $m$ modules of model $\theta$
        $R$: number of global rounds
        $K$: number of local iterations

1: **Input**: global model $\theta$ with $M$ modules
2: **for** $r = 1, \ldots, R$ **do**
3:     Sample clients $S \subseteq \{1, ..., N\}$
4:     **for** each client $i \in S$ in parallel **do**
5:         Initialize local model $\theta_i \leftarrow \theta$
6:         **for** $k = 1, \ldots, K$ **do**
7:             $\theta_i^{1:M} \leftarrow \theta_i^{1:M} - \eta_l \nabla F_i(\theta_i^{1:M})$
8:         **end for**
9:         $\Delta_i \leftarrow \theta_i - \theta$
10:     **end for**
11:     $\theta \leftarrow \theta + \frac{\eta_g}{|S|} \sum_{i \in \mathcal{S}} \Delta_i$
12: **end for**

---

**Algorithm 3** `FedBug`

    **Notation**:
        $\theta^{1:m}$: the first $m$ modules of model $\theta$
        $R$: number of global rounds
        $K$: number of local iterations
        $P$: gradual unfreezing stage percentage

1: **Input**: global model $\theta$ with $M$ modules
2: **for** $r = 1, \ldots, R$ **do**
3:     Sample clients $S \subseteq \{1, ..., N\}$
4:     **for** each client $i \in S$ in parallel **do**
5:         Initialize local model $\theta_i \leftarrow \theta$
6:         **for** $k = 1, \ldots, K$ **do**
7:             $m \leftarrow \min\{M, \lceil \frac{kM}{PK} \rceil\}$
8:             $\theta_i^{1:m} \leftarrow \theta_i^{1:m} - \eta_l \nabla F_i(\theta_i^{1:m})$
9:         **end for**
10:         $\Delta_i \leftarrow \theta_i - \theta$
11:     **end for**
12:     $\theta \leftarrow \theta + \frac{\eta_g}{|S|} \sum_{i \in \mathcal{S}} \Delta_i$
13: **end for**

---