# OpenReview forum: "FedBug: A Bottom-Up Gradual Unfreezing Framework for Federated Learning With Client Drift"
_ICLR.cc/2024/Conference — ICLR 2024 Conference Withdrawn Submission_

### Official Review · Reviewer_3LcF · 2023-10-30

**Soundness:** 2 fair
**Presentation:** 2 fair
**Contribution:** 2 fair
**Rating:** 3
**Confidence:** 4

**Summary:**

This paper studies the problem of client drift in federated learning where due to for example heterogenous data distribution among clients, updates obtained by different clients might affect the global model differently. Specifically, the paper focuses on the image classification applications. To cope with client drift the paper proposes to freeze last layers of a CNN while updating the initial layers. Then unfreeze updating initial layer weights gradually through training. By doing this the author try to control the effect of client drift on the global model. The paper provides some analysis for the performance. And experiments are conducted on image classification datasets.

**Strengths:**

The paper focuses on an important problem of client drift in federated learning. The paper introduces the problem clearly. To show the advantages of the proposed FedBug on image classification tasks, experiments are conducted on various image classification datasets.

**Weaknesses:**

1. The most important weakness of this paper is that it is not clear why the proposed algorithm can effectively deal with client drift. The paper proposes freezing updating a part model weights to control client drift. The most important question here is how can we identify the portion of weights that should be frozen to decrease the client drift. I believe more analysis is needed here to understand what rate of freezing is the best and after how many iterations vanilla averaging update should start. Overall I believe the method proposed by the paper is quite simple and the paper does not provide convincing theoretical analysis to analyze the performance. This limits the contribution of this paper.
2. I think the theoretical analysis provided in the paper belongs to a special case and does not show the effectiveness of the proposed algorithm. Reading the analysis I did not understand the advantages of the proposed FedBug. Specially, the paragraph above definition 1 is not clear to me and I cannot tell that all claims are correct. More importantly, I did not understand what client discrepancy contraction ratio means.
3. The presentation of results in experiments can be improved specially in figures. At first glance it is not clear which bar belongs to the proposed methods. Also experimental results are promising in some cases.

**Questions:**

1. What does client discrepancy contraction ratio show? What are the benefits of smaller client discrepancy contraction ratio?
2. Based on analysis in Theorem 2, the bigger step size the better. Then one can simply chose for example $\eta$=0.99. Is this right? What step size did you use in experiments?

---

### Official Review · Reviewer_Rnja · 2023-10-31

**Soundness:** 2 fair
**Presentation:** 3 good
**Contribution:** 2 fair
**Rating:** 3
**Confidence:** 5

**Summary:**

The paper proposes FedBug, a FL framework designed to mitigate client drift in FL. Unlike standard FL, where clients update all the weights in their model simultaneously, FedBug proposes a sequential training procedure. In FedBug clients first freeze all the weights in their model and then gradually unfreeze layers as they perform training, i.e., layer weights are updated sequentially starting from the input to the output layer. The motivation for this strategy comes from observations that weights in the penultimate and classifier layer exhibit lowest feature similarities among the clients. Theoretical results are provided for a one layer linear network in a synthetic setting. Experiments are conducted on CIFAR-10, CIFAR-100 and Tiny-ImageNet demonstrating the improvement offered by FedBug across a range of training conditions and model architectures.

**Strengths:**

* The paper for the most part is clearly written and does a good job of explaining the central idea.

* FedBug can potentially reduce computation cost at clients while improving overall accuracy, leading to a win-win situation for clients. It also does not add any extra communication cost at clients.


* FedBug is compatible with existing algorithms to tackle client drift such as FedDyn and is experimentally shown to improve performance when combined with these algorithms.

**Weaknesses:**

* I am not very convinced by the motivation for the FedBug strategy. Consider for instance, the initial rounds of training. In this case, the classifier weights would be close to random and therefore the intermediate layers would just be learning features to fit these random weights. Why is this helpful? Wouldn't this slow down training in the initial rounds? I would be interested in seeing how the test accuracy curves for FedAvg and FedBug look across rounds; however I did not find these in the paper (authors just report final accuracies). I would also be more convinced if this approach could be theoretically shown to tackle client drift; however the current theoretical analysis is not too helpful as I discuss below.

* The theoretical analysis is on a very toy setup and not convincing enough to explain the benefit of using FedBug for practical neural network settings . There are too many simplifying assumptions (2 dimensional problem, 2 clients, 1 data point at each client, orthogonal data, 1 layer linear network, $N_{step} = 1$ among others) which significantly limit the generalizability of the results. Focusing on linear networks is okay; however authors have to remove the simplifying assumptions on the data for this to be considered a significant theoretical result. A good reference would be the setting considered in [1] for analysis of linear networks in FL settings.

* Although FedBug can be combined with other algorithms, in order to judge the efficiency of FedBug we have to compare its performance with other algorithms that tackle client drift. It seems that FedBug by itself does not seem to improve performance that much on vanilla FedAvg. In most cases, the performance of (FedAvg + FedBug) is lower than the performance of vanilla FedDyn. This indicates that FedBug is not as successful as tackling client drift as these other algorithm.

**Questions:**

* I was wondering if the authors have considered the effect of tuning the number of local steps, i.e., $K$. It could be that by setting $K$ more carefully, the performance of vanilla FedAvg could match that of (FedAvg + FedBug).


* In Table 1, why is the performance of (FedAvg + FedBug(24%)) so much other better than other baselines for the case of 30% participation, $\alpha = 0.5$? In all other cases, the accuracy of (FedAvg + FedBug(24%)) is lower by almost $1$% compared to the best performing algorithm.


* Again in Table 1, the caption mentions $\alpha \in [0.3, \infty]$; however the table headings mention $\alpha = 0.5$. Please make this consistent.



**References**

[1] Collins, Liam, et al. "Fedavg with fine tuning: Local updates lead to representation learning." Advances in Neural Information Processing Systems 35 (2022): 10572-10586.

---

### Official Review · Reviewer_hW47 · 2023-10-31

**Soundness:** 3 good
**Presentation:** 3 good
**Contribution:** 2 fair
**Rating:** 5
**Confidence:** 4

**Summary:**

The paper proposes a method called FedBUG to mitigate the client drift problem in FL. During the local training, the client starts with a fully frozen model, and then gradually unfreezes parts of the layer during the local training, from the input layer to the output layer. The method borrows the idea of FedBABU, where all layers apart from the classifier are fixed.

**Strengths:**

- The paper is well written and the presentation is good.
- The paper proposed an interesting solution for the client drift and it seems to work well based on the experiments. The method to gradually unfreeze parts of the model can be seen as a creative combination of existing ideas.

**Weaknesses:**

- The definition of GU ratio is not very clear, and the 40% in fig 1 is somewhat confusing. The reader can certainly go into algorithm 1 for more details, but I think the GU ratio can be defined in a more clear and intuitive way.

- Theoretically analysis: (1) the definition of 'client discrepancy' is defined to be the distance between 2 model weights, which can only work for this specific toy example. It is better to provide a definition of 'client discrepancy' in the most general case, and then go into the toy example. (2) it seems that the authors only provide the analysis on the linear models.

- For Table 1, it would be better to provide FedBABU with different aggregation methods to show the superior performance of FedBUG.

- the main part of the paper only provides experiments with 10 clients. With varying numbers of clients, it doesn't provide a comparison with baselines.

- reducing clock time with FedBUG: since the method only gradually unfreezes part of the model, I don't expect a big gain in the compute time. In this case, frozen layers are not much different than the unfrozen ones. The main difference between frozen and unfrozen layers is in terms of memory because for the frozen ones you don't need to store any gradients. Computing gradients with autograd in Pytorch or the equivalent packages is quite cheap and quick nowadays. Also, the compute time itself is not a metric of interest for FL since you need to consider the communication cost, etc. Also, in the Figure, the authors provide the compute time for 10k batches, which I imagine is because the compute time difference is not too much different if the number of batches is small. Under FL cases in a single round, we would rarely have a client that needs to compute 10k batches. Therefore, I think this section doesn't provide significant results and can be misleading.

**Questions:**

- Is there any specific reason for choosing different alpha values for CIFAR10/100 and Tiny-ImageNet?

- FedBABUA is a method to provide a better initial global model that can be able to provide faster/better personalization, so only comparing the global model performance with FedBABU might not seem fair. Have you tried fine-tuning/personalization performance with FedBUG?

---

### Official Review · Reviewer_AHdV · 2023-10-31

**Soundness:** 3 good
**Presentation:** 3 good
**Contribution:** 3 good
**Rating:** 6
**Confidence:** 3

**Summary:**

The paper proposes a new strategy for local training in federated learning called gradual unfreezing. For a hierarchical neural network, the technique freezes the parameters in the top layers and then progressively unfreezes these layers. The paper uses a theoretical toy example to illustrate the rationale of gradual unfreezing. The experiments on CIFAR100 and Tiny-Image net demonstrate the benefits of using the unlearning strategy.

**Strengths:**

The proposed gradual unfreezing technique is intriguing and deserves the attention of the community. The motivations behind the approach are presented with clarity. The inclusion of a theoretical toy example enhances its comprehensibility. Additionally, the numerical analysis provides evidence of the compatibility between gradual unfreezing and popular FL algorithms.

**Weaknesses:**

Though the experiments provide extensive evaluations, I find that the accuracy is not high in general. I understand that the experiments are intended to show GU's power in very difficult scenarios in FL. But it is also desirable to see how GU performs in easy or average scenarios in FL. For example, if the test accuracy is 80% for standard FedAvg, is GU still needed? When is GU needed? Authors can add another set of experiments to show this.

Some minor comments:

1. The following paper is also relevant, but is missing in section 2.
Shi, Naichen, and Raed Al Kontar. "Personalized federated learning via domain adaptation with an application to distributed 3D printing." Technometrics (2023): 1-12.

2. There are some typos in the paper.
In section 4.1, the definition of $L_1$ seems incorrect.

**Questions:**

Do authors provide guidelines on how to choose the unfreezing percentage?